# Fibrosis: Types, Effects, Markers, Mechanisms for Disease Progression, and Its Relation with Oxidative Stress, Immunity, and Inflammation

**DOI:** 10.3390/ijms24044004

**Published:** 2023-02-16

**Authors:** Samar A. Antar, Nada A. Ashour, Mohamed E. Marawan, Ahmed A. Al-Karmalawy

**Affiliations:** 1Department of Pharmacology and Biochemistry, Faculty of Pharmacy, Horus University, New Damietta 34518, Damietta, Egypt; 2Center for Vascular and Heart Research, Fralin Biomedical Research Institute, Virginia Tech, Roanoke, VA 24016, USA; 3Department of Clinical Pharmacology and Toxicology, Faculty of Pharmacy, Tanta University, Tanta 31527, Gharbia, Egypt; 4Pharmaceutical Chemistry Department, Faculty of Pharmacy, Ahram Canadian University, 6th of October City 12566, Giza, Egypt

**Keywords:** fibrosis, inflammation, ECM, TGF-β, organ malfunction, anti-oxidant system

## Abstract

Most chronic inflammatory illnesses include fibrosis as a pathogenic characteristic. Extracellular matrix (ECM) components build up in excess to cause fibrosis or scarring. The fibrotic process finally results in organ malfunction and death if it is severely progressive. Fibrosis affects nearly all tissues of the body. The fibrosis process is associated with chronic inflammation, metabolic homeostasis, and transforming growth factor-β1 (TGF-β1) signaling, where the balance between the oxidant and antioxidant systems appears to be a key modulator in managing these processes. Virtually every organ system, including the lungs, heart, kidney, and liver, can be affected by fibrosis, which is characterized as an excessive accumulation of connective tissue components. Organ malfunction is frequently caused by fibrotic tissue remodeling, which is also frequently linked to high morbidity and mortality. Up to 45% of all fatalities in the industrialized world are caused by fibrosis, which can damage any organ. Long believed to be persistently progressing and irreversible, fibrosis has now been revealed to be a very dynamic process by preclinical models and clinical studies in a variety of organ systems. The pathways from tissue damage to inflammation, fibrosis, and/or malfunction are the main topics of this review. Furthermore, the fibrosis of different organs with their effects was discussed. Finally, we highlight many of the principal mechanisms of fibrosis. These pathways could be considered as promising targets for the development of potential therapies for a variety of important human diseases.

## 1. Introduction

Fibrosis is a condition that develops slowly but eventually leads to tissue degeneration, which has devastating consequences for heart, lung, liver, kidney, and skin disorders [1]. It occurs when there is an excessive buildup of fibrous connective tissue in the extracellular matrix (ECM) area of tissues that have been injured. The basic components of fibrotic scar tissue and a mixture of fibrotic cells are collagens, especially types I and III. Extremely fibrotic tissue will experience persistent healing issues, which will lead to organ or tissue malfunction [2]. Pro-fibrotic cells and pro-fibrotic factors and cytokines (growth factors/cytokines) are among the mediators of fibrosis in various tissues, along with other variables including the ECM, tissue vascular injury, mechanical tension, and oxidative stress [3]. In tissue trauma and fibrogenic disorders, myofibroblasts or myofibroblast-like cells are the primary cell type contributing to the collagen synthesis and turnover of the ECM [4]. One of the markers that smooth muscle cells all share is alpha-smooth muscle actin (α-SMA), which is expressed by myofibroblasts. Mesenchymal cells are widely distributed throughout the body and include the bone marrow cells that are undergoing epithelial to mesenchymal change as a result of damage and fibroblasts. By way of the bloodstream, they are typically attracted to the sites of injury as monocytes, which are then induced to develop into macrophages by inflammatory and profibrotic cytokines [5]. The myofibroblast is a key player in the emergence of fibrotic diseases, as shown in Figure 1. The myofibroblast has a role in several processes, such as organ fibrosis, embryologic development, and the stromal response to epithelial malignancies [6]. Recent discoveries that myofibroblasts derive from a variety of cellular origins depending on the normal or pathological condition support this hypothesis. Progressive fibrotic illness can be brought on by a variety of different factors. Minor human leukocyte antigen abnormalities in transplants, high serum cholesterol, myocardial infarction, poorly controlled diabetes, obesity, and hypertension are a few examples of conditions that might cause chronic inflammation [7]. It is now understood that a variety of factors, including immunological response and inflammation, affect how fibroblasts differentiate and become activated.

## 2. Markers of Fibrosis and Their Roles

### 2.1. TGF-β’s Function in Mediating Fibrosis

TGF-β1 is a multifunctional regulator that affects a wide range of basic biological processes, such as cell division, proliferation, and apoptosis, as well as embryonic development and tissue homeostasis [8]. Three TGF-β isoforms found in mammals—transforming growth factor type 1 (TGF-β1), transforming growth factor type 2 (TGF-β2), and transforming growth factor type 3 (TGF-β3), share 70–82% amino acid similarity and have similar functions in many systems [9]. TGF-β3 is mostly found in mesenchymal cells, whereas TGF-β1 is expressed in endothelial, hematopoietic, and connective tissue cells. TGF-β2 is expressed in epithelial, neuronal, and connective tissue cells. In fact, it can downregulate cell junctions favoring cell proliferation and survival in inflammatory diseases [10,11]. The most common isoform that can be produced by all different types of renal resident cells is TGF-β1 [12]. Following synthesis, latency-associated peptide (LAP) and the latent form of TGF-β1, which binds to the Latent TGF-β binding protein (LTBP) in the target tissues, are released [13]. Numerous stimuli, such as reactive oxygen species (ROS), can cause TGF-β1 to be released from the LAP and LTBP and be activated. TGF-β1 will exhibit its potent biological activity after it is released from the LAP/LTBP protein complex, as shown in Figure 2. Transforming Growth Factor-Beta (TGF-β) is a diverse regulatory and fibrogenic protein with 3 isoforms, and TGF-β1 is the most common, followed by TGF-β2 and TGF-β3 [14]. These three separate TGF-β isoforms (TGF-β1, TGF-β2, and TGF-β3) have been found in mammals and share 70–82% homology in the amino acids. All 3 proteins are secreted in a latent complex and dimerized, and the complex contains a latent TGF-β binding protein (LTGF-β). This latent complex may bind the ECM, where TGF-β may be stored until activated. By cleaving LTGF-β, thrombin and other serum proteases activate TGF-β. The three isoforms of TGF-β are encoded by three independent genes—TGF-β1, TGF-β2, and TGF-β3, respectively. Each of these three genes has a complex promoter that contains both negative and positive regulatory elements and is receptive to a wide range of repressors and activators. One study reported that TGF-β1 and TGF-β3 are regulated differently by showing instances where TGF-β1 and TGF-β2 caused different reactions in a specific cell. Furthermore, studies have shown TGF-β isoforms controlling the expression of other modulators and receptors compared to studies of regulators of the transcription of the TGF-β isoforms themselves, and our limited understand of regulation at this level. There are 3 TGF-β receptors (TGF-βR1, TGF-βR2, and TGF-βR3), and all three isoforms of TGF-β signal via these three receptors. In vitro data suggest that TGF β isoforms act through the same receptors and also the same SMAD 2/3 pathway, but they have some distinct effects in regulating fibrosis that are only superficially understood—with TGF-β1 and TGF-β2 having a primarily pro-fibrotic effect and, conversely, TGF-β3 having more of an anti-fibrotic effect [15]. 

TGF-β1’s active form is a dimer that is often enhanced by an inter-subunit disulfide bond and stabilized by hydrophobic contacts. Specific TGF-β1 receptors (TBR), such as TGF-β1 receptors type I (TBRI), type II (TRBII), and type III (TRBIII), bind to activated TGF-β1 to begin intracellular signaling. TGF-β1/Smad signaling activates receptor-associated Smads (R-Smads), such as Smad2 and Smad3, as shown in Figure 3. The target genes’ transcription is then affected by the translocation of phosphorylated Smad2 and Smad3 into the nucleus. Interestingly, Smad3 can influence the induction of Smad7, an inhibitory Smad [16]. To significantly impact TGF-/Smad signaling, the R-Smads and Smad7 engage in competition for binding to the activated receptors. There is strong evidence to support the claim that TGF-β is necessary for the onset of fibrosis, including the following points: fibrotic disorders affect nearly all organ systems in humans, and experimental animal models of fibrosis show the increased expression of TGF-β mRNA and/or protein. In addition, multiple organs and tissues in transgenic mice that overexpress activated TGF-β globally develop fibrosis. By blocking activated TGF-β via anti-TGF-β antibodies, TGF-β binding proteins, TGF-β type 1 receptor, or by overexpressing the dominant-negative TGF-β type 2 receptor, fibrosis is reduced in experimental animals [17].

### 2.2. Role of Smads in Fibrosis

Numerous studies have demonstrated that the disruption of the TGF-β1/Smad pathway was an important pathogenic element in tissue fibrosis [18]. It is known that a key mechanism of TGF-β signaling in progressive fibrosis is Smad signaling. Receptor-regulated Smad (R-Smad), common partner-Smad (co-Smad), and inhibitory Smad (I-Smad) are the three primary subtypes of Smad signaling [19], as illustrated in Figure 4. In patients with chronic kidney disease (CKD) and the fibrotic kidneys of animal models, Smad2 and Smad3 are highly active [20]. Obstructive nephropathy, diabetic nephropathy, hypertensive nephropathy, and drug-toxicity-related nephropathy are inhibited by the knockout of the Smad3 gene. Notably, Smad3 inhibits ECM breakdown by directly binding to the promoter region of collagens to activate their production, promoting renal fibrosis [21]. In addition, Smad4 plays a crucial role in the nucleocytoplasmic transfer of the Smad2/3 and Smad1/5/8 complexes and functions as a common Smad for TGF-β/BMP signaling [22]. Studies have found that the specific deletion of Smad4 from renal tubular epithelial cells attenuates unilateral ureteral obstruction-induced renal fibrosis by suppressing Smad3 responsive promoter activity and reducing the binding of Smad3 to the target genes [23]. 

The loss of Smad4 in mesangial cells inhibits TGF-β1-induced ECM deposition [24]. Smad3 is required for the induction of Smad7, an inhibitory regulator in the TGF-β/Smad signaling pathway. Smad3’s negative feedback loop thereby prevents TGF-β1/Smad signal transduction [25]. Additionally, Smad7’s efficient control mechanism on TGF-β signaling occurs. TGF-β not only stimulates Smad7 transcription, but also encourages Smad7 breakdown by turning on Smad3. Most importantly, the discovery that deletion of Smad7 increases renal fibrogenesis in diabetic nephropathy, obstructive nephropathy, and hypertensive nephropathy further defines the functional role of Smad7, suggesting Smad7 as a therapeutic agent for the treatment of CKD [26]. Overall, strong evidence suggests that a fundamental characteristic of renal fibrotic disorders is the hyperactivation of Smad3 in conjunction with the increasing degradation of Smad7. More significantly, it was shown that one of the key pathways for mediating the fibrotic response was an imbalance of Smad3 and Smad7 [27]. In this regard, downregulating Smad3 and simultaneously upregulating Smad7 appears to be an effective method for treating fibrosis by rebalancing the aberrant Smad3/Smad7 ratio. Additionally, the accumulation and activation of myofibroblasts, the overproduction of ECM, and the reduction in ECM degradation are caused by the equilibrium changes between Smad3 and Smad7. Additionally, Smad3 causes fibrosis by upregulating miR-21 and miR-192, while downregulating miR-29 and miR-200 [28].

### 2.3. Role of Fibroblast Growth Factor23 (FGF23) in Fibrosis

FGF23 is a 32 kDa proteohormone that belongs to the fibroblast growth factor (FGF) family [29]. The newly identified endocrine hormone FGF23 is produced by osteoblasts and osteoclasts in bone and affects the kidney and parathyroid glands to control phosphate homeostasis and vitamin D metabolism [30]. High levels of FGF23 have also been proven to have pathologic effects in addition to its physiological benefits. Studies have demonstrated that FGF23 can directly cause left ventricular hypertrophy and is considerably higher in both chronic renal disease patients and the general population [31]. Through the stimulation of β-catenin, FGF23 aids in the promotion of myocardial fibrosis [32]. Studies revealed that in an ischemia situation, proliferation, the synthesis of collagen I and III, and β-catenin activation are all promoted by FGF23. Significantly, the endogenous cardiac FGF23 activates paracrine signaling pathways that promote myocardial fibrosis during myocardial infarction or ischemia-reperfusion. Furthermore, FGF23-induced profibrotic interactions between cardiac myocytes and fibroblasts accelerated heart fibrosis [33]. FGF23 synergistically activated fibroblasts in the presence of TGF-β1, which is known to promote myofibroblast trans-differentiation via the Smad-3 and Wnt signaling pathways, as opposed to acting as a profibrotic on its own [34].

### 2.4. Role of Connective Tissue Growth Factor (CTGF) in Fibrosis

A member of a small family of proteins, CTGF has three to four domains that are homologous to other proteins and a highly conserved disulfide bonding structure [35]. CTGF plays a significant biological role. Numerous signaling pathways that regulate angiogenesis, myofibroblast activation, extracellular matrix deposition, and remodeling, all of which contribute to tissue remodeling and fibrosis, are altered. Many cytokines and pathophysiological circumstances can cause the production of CTGF. Its presence causes other cells, such as epithelial cells via epithelial-mesenchymal transition (EMT), stellate cells, resident fibroblasts, or fibrocytes to transdifferentiate into myofibroblasts. CTGF stimulates the ECM proteins that the myofibroblasts deposit and change, and it also stimulates the myofibroblasts’ activation [36]. Fibrosis and tissue remodeling are the results of this. Localized hypertension may result from tissue remodeling in the vasculature, which may subsequently cause CTGF production, setting off a positive feedback loop that encourages more tissue remodeling. As a result, the expression of CTGF can be involved in several positive feedback loops that may contribute to the progressive nature of fibrosis [37]. 

According to certain reports, TGF-β and CTGF work together to encourage chronic fibrosis [38]. TGF-β is directly bound by CTGF, which enhances its reactions, as demonstrated in Figure 5. The method is based on CTGF’s chaperone activity, which enhances TGF-β’s affinity for its various receptors and results in stronger and longer-lasting responses. The TGF-β inducible early gene (TIEG-1) transcription factor is activated during the endogenous synthesis of CTGF by TGF-β, which also suppresses SMAD-7’s transcription. TGF-β prevents SMAD-7’s feedback control, maintaining the activation of TGF-β signaling. This may be important for pathological diseases where CTGF expression is elevated [39]. The TGF-β1/Smad signaling route and angiotensin II (Ang II) are known to promote CTGF expression through the AT1-extracellular signal-regulated kinase/p38 mitogen-activated protein kinase pathway. The production of Ang II, which increases the expression of other factors such as TGF-β, CTGF, plasminogen activator inhibitor-1 (PAI1), and nuclear factor kappa B (NF-κB), causes many biological effects to be activated and amplified. This leads to the initial recruitment of neutrophils, which are then replaced by macrophages and T lymphocytes, inducing an immune response that results in interstitial nephritis [40]. 

### 2.5. Role of Nuclear Erythroid 2-Related Factor 2 (Nrf2) in Fibrosis

Nrf2 has a considerable impact on the regulation of several anti-fibrotic substances or pathways [41]. Nrf2 is typically made in the cytoplasm, where kelch-like ECH-associated protein1 (KEAP1) also mediates its ubiquitination and breakdown, regulating Nrf2 through negative feedback. This signalling is also involved in cancer progression and chemoresistance. Moreover, modulating the expression of important antioxidant enzymes plays a key role in cancer prevention [42,43,44]. It is also referred to as an Nrf2 inhibitor because it can sense the redox state and negatively regulate the activity of Nrf2. To combat oxidative damage, the main cause of fibrosis, the expression of certain antioxidant genes is typically elevated among other anti-fibrotic proteins [45]. Fibrotic illnesses eventually show the harmful nature of ectopic collagen accumulation after a period of oxidant damage. The matrix metalloproteinase/Tissue inhibitors (MMP/TIMP) system, which is induced by a few cytokines present in the fibrosis milieu, such as TGF-β, may result in the degradation of the native extracellular matrix. The increased production and ectopic storage of collagen, which are common clinical characteristics of several fibrotic disorders, are caused by the TGF-β/SMADs pathway [46]. Additionally, the extracellular matrix’s native components may be broken down by the aberrant activation of the MMP/TIMP system brought on by TGF-β, creating space for ectopic collage accumulation. Nrf2 can block the TGF-β/SMADs pathway, which reduces collagen synthesis and lessens fibrosis [47]. 

### 2.6. Role of Renin-Angiotensin-Aldosterone System (RAAS) in Fibrosis

Although Ang II appears to be the primary hormone promoting cardiac fibrosis in hypertensive heart disease, the RAAS system demonstrates profibrotic activity [48]. According to research, Ang II stimulates collagen-secreting myofibroblasts via increasing fibroblast proliferation and differentiation, generating TGF-β1, and directly activating NADPH oxidase activity, as demonstrated in Figure 6. Furthermore, Ang II also improves TGF-β1 signaling by raising Smad2 levels and promoting phosphorylated Smad3’s nuclear translocation [49]. TGF-β1 in turn increases the cardiac myofibroblasts’ production of proteoglycans, fibronectin, and interstitial collagens. It also encourages myofibroblasts to initiate the production of an autocrine cycle of myofibroblast activation and differentiation. According to studies, transgenic mice that overexpress TGF-β1 develop hypertrophic cardiac myocytes and interstitial fibrosis, which are both signs of cardiac hypertrophy. Treatments that target the RAAS or TGF-1 pathways may be helpful to slow the progression of fibrosis in progressive renal disease, hypertensive cardiac disease, and hepatic fibrosis [50].

## 3. Oxidative Stress and Fibrosis

An imbalance between the production of ROS, reactive nitrogen species (RNS), and antioxidant defense leads to oxidative stress, which results in cellular dysfunction and tissue damage [51,52,53]. Highly reactive oxygen metabolites include hydrogen peroxide (H_2_O_2_), superoxide anion (O_2_•), hydroxyl radical (HO•), Nitric oxide (NO), and oxygen (O_2_), which react to form nitric oxide (NO) and its derivatives, such as peroxynitrite. All molecular, cellular, and tissue problems brought on by excessive ROS generation and/or depleted antioxidant defenses are collectively referred to as “oxidative stress” [54]. It is crucial for inducing fibrosis that oxidative stress and TGF-β interact. A positive feedback loop is created when TGF-β increases the creation of ROS, which causes oxidative stress, and when oxidative stress activates latent TGF-β, as demonstrated in Figure 7. Additionally, peroxynitrite and other RNS can be created when superoxide and NO combine. The inducible nitric oxide synthase (iNOS) mostly produces NO in the lung, particularly during inflammation [55]. Human lung cells extensively exhibit constitutive forms of NOS, which help to increase NO generation. Numerous genes involved in cell proliferation, cell death, and fibroblast survival can be activated by these oxidants. It appears that ROS produced by critical target cells’ mitochondria plays a crucial role in mediating fibrosis. ROS (such as H_2_O_2_ and O_2_) are produced as a result of mitochondrial malfunction because the electron transport chain becomes decoupled from proton pumping and releases ROS into the cytosol [56]. When produced, typically by activated inflammatory cells, and additionally by non-inflammatory ones, the NOX family of oxidoreductases catalyzes one or two electron reductions to form O_2_ and H_2_O_2_, which aid in cellular communication, the killing of invasive microorganisms, and harm to the surrounding host tissues. The key function of NOX, particularly isoforms NOX1, NOX2, and NOX4, in the etiology of pulmonary fibrosis has been demonstrated by recent investigations [57]. TGF-β1 induces NOX4, which then encourages crucial developments in fibrotic lung illness. Studies have shown that bleomycin-induced fibrosis in mice can be prevented by genetically impairing NOX4 or by pharmacologically inhibiting it [58]. The connection of fibrosis with ROS and oxidative stress implies that supplementation with nutrients or diets with antioxidants will, in addition to disease-specific therapies and the inhibition of TGF-β signalling, be beneficial. Resveratrol, a naturally occurring antioxidant found in grapes and certain berries, has been advocated as a treatment for lung fibrosis, chronic obstructive pulmonary disease, and respiratory conditions such as asthma [59]. Animal studies have demonstrated that both catalytic and scavenger antioxidants can reduce lung fibrosis brought on by bleomycin. In rats and mice, bleomycin-induced lung fibrosis is lessened when SOD is delivered liposomally or through a lecithin-based carrier with or without catalase. The bleomycin-induced lung fibrosis in mice is reduced by the catalytic antioxidant porphyrin MnTBAP. Although additionally demonstrated to have protective benefits, a deficiency in vitamin E amplifies the pulmonary fibrosis caused by bleomycin in rats. The thiol-containing antioxidant that has been studied the most is NAC. In rats and mice, lung fibrosis is reduced by oral and inhaled NAC. NAC has also been demonstrated to reduce bleomycin-induced NF-κB activation and to restore the redox equilibrium of lung GSH. It has also been demonstrated that a number of prodrugs, including erdosteine and amifostine, which generate active thiol-containing metabolites, reduce bleomycin-induced lung fibrosis in animals. Lazaroids also protect rats’ lungs from developing fibrosis brought on by bleomycin [60]. In rats given bleomycin treatment, it has been discovered that several naturally occurring products containing polyphenolic chemicals, such as Ginkgo biloba extracts and curcumin, decrease lung oxidative stress and fibrosis [61].

## 4. Antioxidant

Antioxidants have been defined in many different ways, and in a very broad sense, they are agents that decrease steady-state ROS levels and protect cellular macromolecules from oxidative modification [53,62]. A classic antioxidant is an agent that can rapidly react with ROS, producing less-reactive species. Regardless of the mechanisms, antioxidants decrease oxidative stress and restore redox balance in biological systems [63], as shown in (Table 1). A growing number of research articles suggest that exogenous antioxidant supplementation could be an approach worth considering for the future treatment of Idiopathic pulmonary fibrosis (IPF) [64]. Antioxidant supplements could complement the inadequately working lung antioxidant defense system and reduce oxidative stress, also acting as anti-inflammatory agents. Antioxidant activity is exhibited by both food-derived antioxidant compounds and drugs. In-vivo studies have demonstrated the effectiveness of the antioxidant drug *N*-acetylcysteine (NAC) in preserving vital functions of the lung in IPF patients when used together with standard treatment [65]. A similar conclusion, that is, that combined antioxidant therapy was safer and more effective than monotherapy, was drawn by Kandhare et al., who performed a meta-analysis on antioxidant treatment (NAC and lecithinized superoxide dismutase) in IPF patients [66].

## 5. Inflammation and Fibrosis

Regeneration and fibrosis are both significantly triggered by tissue injury and inflammation. The activation of a range of distinct innate and adaptive immune system cell types by tissue injury regulates the type and orientation of inflammation in addition to causing general inflammation [75,76]. As soon as the wound resolves on its completion of repair and recovery, the recruitment of inflammatory cells to the site of injury in wound healing is a crucial step. As a result, inflammation frequently occurs before fibrosis [77]. Premier inflammation is brought on by endocytosis and phagocytosis mediated by cytokines. The first cells to be recruited and activated are neutrophils. Additionally, the primary event that initiates fibrogenesis and promotes the release of fibrogenic cytokines is the activation of T cells. Mice lacking mature B and T cells are protected from developing fibrosis following obstructive damage [78]. In a summary, these results confirm the crucial part lymphocytes play in the start of fibrosis. Macrophages invade injured tissues and release fibrogenic cytokines once neutrophils and T cells are activated. In fibrotic tissues, the macrophage is a key generator of TGF-β1 [79]. The NF-κB is strengthened when interferon-ɣ (IFN-ɣ) activates macrophages. Additionally, macrophages exhibit a characteristic pro-inflammatory phenotype, create a wide range of chemokines as well as ROS, and contribute to tissue damage and fibrosis in a pathogenic manner [80]. Therefore, macrophage depletion reduces fibrosis following injuries, while macrophage recruitment increases the fibrotic lesions. As a result, the initiation and progression of the fibrotic disease may be caused by the infiltration and activation of inflammatory cells. There is growing evidence that inflammation, particularly chronic inflammation, is closely related to fibrosis and plays a significant role in the onset and progression of renal disease, diabetes, cancer, and heart disease [81]. Fibrosis is marked by an increase in the inflammatory response, tissue destruction, and the release of numerous inflammatory cytokines, including TGF-β1, tumor necrosis factor-alpha (TNF-α), monocyte chemoattractant protein (MCP-1), Interleukin-6 (IL-6), and IL-8. This results in a pro-inflammatory microenvironment that amplifies tissue injury [82]. EMT, which is becoming more and more well-documented as an essential component of tissue fibrogenesis following renal damage, is a process by which differentiated epithelial cells give rise to matrix-producing fibroblasts and myofibroblasts. Podocyte dysfunction, proteinuria, and glomerulosclerosis may all be caused by EMT, which happens in glomerular disorders. Mesenchymal cells called fibroblasts play a role in the development of renal illness. The renal illness causes an increase of fibroblasts, which can be activated by many cytokines, particularly TGF-1, or can develop into myofibroblasts, as demonstrated in Figure 8.

### 5.1. Role of TNF-α in Fibrosis

TNF-α, a pleiotropic cytokine produced by various immune cells including macrophages and monocytes, is one of these cells. Several pathways involved in apoptosis, proliferation, and inflammation can be activated by TNF-α. [83]. The pathogenesis of the persistent inflammation that results in fibrosis has been linked to TNF-α. Since the information currently available indicates that TNF-α regulates collagen synthesis in a tissue-specific manner, the role of TNF-α in inflammation-induced fibrosis appears to be complex [84]. Reduced experimentally induced lung fibrosis or renal fibrosis is seen in mice lacking either the 55-kDa (TNFR1) or 75-kDa (TNFR2) TNF receptors [85]. Despite equal increases in type I collagen gene expression as in wild-type mice, TNF receptor null animals had lessened pulmonary fibrosis, which was accompanied by a diminished induction of tissue inhibitor of metalloproteinase-1 (TIMP-1) mRNA. TIMP-1 is an inhibitor of matrix metalloproteinases (MMPs), which break down collagen by cleaving different parts of the extracellular matrix. These results in TNFR-deficient mice imply that TNF typically participates in the up-regulation of TIMP-1 upon lung injury and may facilitate fibrosis by preventing collagen breakdown [86]. 

### 5.2. Fibrosis and the NF-κB Pathway

NF-κB is a crucial transcription factor in the regulation of inflammation, the immune system, and cancer [87]. In such circumstances, NF-κB controls the transcription of specific genes that act as significant regulators or effectors of the host’s responses to intracellular and extracellular stressors. NF-κB family transcription factors can activate cells, leading to tissue fibrosis [88]. NF-κB can stimulate Ang II by binding to the AT1 and AT2 receptors. The NF-κB family of transcription factors has several potential correlations, and Ang II likely stimulates various NF-κB isotypes at various stages of the development of renal disease. Since the tissue enzyme transglutaminase, which is a protein expressed by kidney tubular epithelial cells, activates latent TGF-β1 and is controlled by NF-κB, this fact appears to be linked to an increase in renal fibrosis [89]. By increasing α-SMA synthesis, fibroblasts are encouraged to become myofibroblasts. It seems that during liver fibrosis, NF-κB may contribute to a rise in α-SMA expression [90]. TNF-α production may be stimulated by Ang II, and TNF-α production may in turn increase NF-κB activation. Studies reported that in mouse lungs, multi-walled carbon nanotubes (MWCNT) activate NF-κB in fibroblasts and myofibroblasts, promoting the production of osteopontin (OPN) and MWCNT-induced lung fibrosis [91].

### 5.3. Role of Jun N-Terminal Kinase (JNK) in Fibrosis

Both leukocytes and non-leukocytes can be stimulated by the JNK pathway to produce an inflammatory response. Depending on the type of cell and the degree of tissue injury, the stimuli that cause JNK activation may take on different forms [92]. The transcriptional regulator AP-1 is a crucial mechanism by which the JNK pathway promotes inflammation, as demonstrated in Figure 9. JNK can phosphorylate c-Jun, which enables it to dimerize with c-Fos to produce AP-1, which subsequently controls many genes that govern the inflammatory response, including cytokines (such as TNF-α) and chemokines [93]. JNK signaling in tubular epithelial cells can be regarded as a crucial determinant in the development of tubulointerstitial damage and fibrosis based on the frequent observation of tubular JNK activation in human and experimental kidney disease [94]. This may cause the stimulation of pro-inflammatory and pro-fibrotic reactions, cell death by apoptosis or necrosis, and the dedifferentiation of tubular cells, which activates a gene profile associated with the mesenchymal cell type. Interstitial fibrosis, macrophage infiltration, deteriorating renal function, and tubular injury were all significantly correlated with JNK activation in tubulointerstitial cells [95]. In response to numerous stresses, tubular epithelial cells frequently activate JNK. Using unilateral ureteric obstruction, the function of JNK signaling in renal interstitial fibrosis was initially investigated. One of the main sites of JNK activation in kidney illness is tubular epithelial cells, and systemic JNK inhibition shows that this pathway plays a pathogenic role in models of tubulointerstitial damage and fibrosis [96]. TNF-α mediated damage has been linked to the JNK pathway [97]. In the myofibroblasts of human fibrotic/cirrhotic livers, JNK activation was seen. In primary HSCs, JNK1 disruption prevented them from transdifferentiating into myofibroblasts, but resulted in cell death. JNK1 and JNK2 upregulate α-SMA levels in HSCs under non-stressful conditions, but only JNK1 participates in α-SMA upregulation under stress conditions induced by TGF-β during liver fibrosis [98]. 

## 6. Fibrosis Is Mediated via the Janus Kinase (JAK)-Signal Transducer and Activator of Transcription (STAT) Pathways

JAKs are receptor-associated tyrosine kinases that play crucial parts in the signaling of cytokines and growth factors. The kinases become autophosphorylated and activated as a result of cytokine binding to the receptor, which recruits and phosphorylates STAT proteins to activate them. Target gene transcription is aided by the dimerization, nuclear translocation, and phosphorylation of STATs [99]. JAK-STAT signaling is a well-known and important mediator of inflammation, and small-molecule JAK inhibitors have received clinical approval for the treatment of rheumatoid arthritis and are currently being considered for additional inflammatory illnesses [100]. Normal fibroblasts are stimulated by TGF-, which results in the phosphorylation of JAK2, the activation of STAT3, and the transcription of collagen [101]. Notably, the research by Chakraborty et al. indicates that JAK2 and the kinases JNK are both responsible for phosphorylating STAT3 (Figure 10). In vitro, STAT3 depletion or pharmacologic suppression blocks TGF-β induced fibroblast activation and collagen release. Additionally, therapy with a specific STAT3 inhibitor or conditional STAT3 knockdown in fibroblasts reduces experimental fibrosis in various skin fibrosis models [102].

## 7. Role of Wnt Pathway in Fibrosis

Wnt proteins are secreted ligands that engage with Frizzled receptors and co-receptors for low-density lipoprotein receptor-related proteins (LRP5/6) to cross the plasma membrane and deliver their signal [103]. When Wnt proteins attach to their receptors, a series of intracellular signaling processes are triggered that involve the proteins Dishevelled, Axin, Adenomatosis Polyposis Coli, and Glycogen Synthase Kinase-3, and end with the stabilization of β-catenin. β-catenin translocates to the nucleus, as demonstrated in Figure 11, where it interacts with T-cell factor/lymphoid enhancer-binding factor (Tcf/Lef), which causes the transcription of Wnt target genes [104]. Numerous illnesses have been linked to the abnormal stimulation of the canonical Wnt pathway, which can be caused by mutations in intracellular regulators, changes in the production of Wnt proteins, or endogenous inhibitors of Wnt signaling. Growing evidence suggests that canonical Wnt signaling activation may play a significant role in fibrogenesis. The pathogenesis of pulmonary, renal, cutaneous, and hepatic fibrosis as well as scarring after cardiac fibrosis and fibrosis following muscular dystrophy have all been linked to pathologically activated canonical Wnt [105]. The two types of Wnt signaling are β-catenin-dependent (“canonical” Wnt signaling) and β-catenin-independent (“non-canonical” WNT signaling). Canonical Wnt signaling is adequate and necessary for fibrotic tissue remodeling and is active in a variety of fibrotic disorders in many organs [106]. The overexpression of Wnt proteins, the downregulation of endogenous inhibitors, and the nuclear accumulation of β-catenin are thought to be the causes of the enhanced activation. With nuclear translocation of β-catenin and enhanced target gene transcription, TGF-β may trigger canonical Wnt signaling in skin fibroblasts. In fibrosis, the canonical Wnt pathway is crucial for the stimulation of fibroblasts and the release of collagen. Wnt signaling accelerated the release of extracellular matrix components, promoted fibrosis, and triggered myofibroblast development from resting fibroblasts [107]. It should be noted that Wnt activation-induced fibrotic illness was more significant than the impacts of other fibrotic pathways. The canonical Wnt pathway’s activation and the strong profibrotic effects it has led some researchers to hypothesize that the Wnt pathway might be a potential target for novel antifibrotic approaches [108]. 

## 8. Immunity and Fibrosis 

Both innate and adaptive immunity contribute to fibrogenesis [109]. Based on the differential expression of surface molecules and cell functions, T cells can be divided into CD8+ cytotoxic T lymphocytes (CTL), natural killer T (NKT) cells, T helper cells including Th1, Th2, Th9, Th17, Th22, and T follicular helper (Tfh) cells and regulatory T (Treg) cells [110]. Numerous studies have confirmed that the immune response plays a significant role in fibrosis and fibrotic diseases. Notably, the systematic identification of immune cells and signaling pathways is still necessary for the development of novel therapeutics.

### 8.1. T Cell Function and Fibrosis Pathways 

#### 8.1.1. Fibrosis and Th1 Cells

One of the key steps leading to fibrosis is inflammation. Naive CD4 cells are converted into Th1 cells by IL-12, which releases the pro-inflammatory cytokine IFN-γ [111]. IFN-γ reduces fibrosis by inhibiting collagen synthesis that is produced by fibroblasts [112]. In order to break down ECM components, IFN production increases the expression of matrix metalloproteinases (MMPs), such as MMP-2, MMP-7, MMP-9, and MMP-13. This proteolytic activity diminishes fibrosis and modifies ECM remodeling [113].

#### 8.1.2. Fibrosis and Th9 Cells

Infections with parasites and allergic reactions led to the discovery of Th9 cells. The pathological processes of many diseases, including inflammatory disorders, viral diseases, autoimmune diseases, and cancer are influenced by the pleiotropic cytokine IL-9 [114]. Target cells that are stimulated include dendritic cells, mast cells, and CD8+ T lymphocytes. Patients with perivascular fibrosis due to Schistosoma mansoni infection have shown elevated serum IL-9 levels. IPF fibrosis patients as well as a rat model of silica-induced lung fibrosis were both revealed to have considerably increased IL-9 levels [115]. IL-9 is also markedly elevated in liver cirrhosis patients and is crucial to the development of hepatic fibrosis. Another study found that IL-9 activated the Raf/MEK/ERK signaling pathway signaling model of liver fibrosis brought on by the chemical carbon tetrachloride (CCl_4_) [116]. 

#### 8.1.3. Fibrosis and Cytotoxic T Cells (CTLs, CD8+ T Cells) 

Killing tumor cells and infected cells requires the activity of cytotoxic T lymphocytes (CTLs, CD8+ T cells), which express the CD8 glycoprotein as an identity marker. They also play critical roles in a variety of conditions connected to fibrosis [117]. In a mouse model of severe cerebral ischemia, CD8+ T cells invaded the perivascular space and released the cytokine IL-16 to attract monocytes and CD4+ T cells, which decreased the fibrosis of the muscles. Additionally, CD8+ T cells and IFN-γ decreased the CD4+ T cell-induced monocyte-to-fibroblast transition in a renal fibrosis paradigm [118]. TNF can be produced by activated CD8+ T lymphocytes, which also cause thyroid fibrosis. In an IL-21-dependent way, they can also release IL-13 to facilitate bleomycin-induced lung fibrogenesis. 

#### 8.1.4. Tfh Cells and Fibrosis

T follicular helper (Tfh) cells, which are recognized by their expression of the lineage-specific transcription factor Bcl6 and the secretion of IL-21, are necessary for B cell activity [119]. Macrophages are responsible for driving the differentiation of Tfh cells after Schistosoma japonicum infection. The infiltrating Tfh cells then promote the development of hepatic granulomas and cause severe liver fibrosis [120]. Patients with primary biliary cirrhosis have higher amounts of these cells as well (PBC). The peripheral blood contains more CXCR5 + ICOS +PD-1 + Tfh cells than usual. Furthermore, systemic sclerosis patients’ dermal fibrosis is closely correlated with the presence of CXCR5 + ICOS + PD-1 + Tfh cells (SSc). A mouse model of sclerodermatous GVHD (GVHD-SSc) revealed that IL-21 and MMP-12 depend on the number of these profibrotic Tfh cells [121]. Additionally, administering ICOS antibodies and IL-21 can both significantly decrease skin fibrosis. Tfh cells, a new subpopulation of T cells, may offer fresh perspectives on treatments for fibrotic illness [122].

## 9. Effect of Fibrosis on Different Organs

### 9.1. Cardiac Fibrosis

Cardiac fibrosis is an aberrant thickening of the heart valves caused by unsuitable fibroblast proliferation, but it is more frequently used to describe the excessive deposition of ECM in the cardiac muscle, as described in Figure 12 [123]. Numerous cell types, including cardiomyocytes and fibroblasts, can be found in the myocardium. About two-thirds of the heart’s cells are fibroblasts, which are located in the connective tissue. They play a role in a variety of cardiac functions, such as controlling the ECM’s equilibrium, rebuilding the ECM, and producing growth factors and cytokines [124]. The typical role of fibrocyte cells is to secrete collagen and support the heart’s structural integrity. When this process is overactive, it results in the valve’s thickening and fibrosis, which can induce valvular malfunction and heart failure. After an acute myocardial infarction, a high number of cardiomyocytes suddenly disappear, which sets off an inflammatory response and eventually causes the dead myocardium to be replaced with a collagen-based scar. Various pathophysiologic circumstances cause more hidden collagen-like deposition; both ventricular dilatation and combined diastolic and systolic heart failure can be brought on by pressure overload brought on by hypertension or aortic stenosis as well as aging, which is connected to progressive cardiac fibrosis that leads to diastolic heart failure in elderly people [125]. Additionally, both in experimental models and human patients, a variety of toxic insults like alcohol or anthracyclines, as well as metabolic abnormalities such as diabetes and obesity, cause progressive fibrotic alterations in the myocardium [126]. It is a key factor in the majority of cardiac pathologic diseases and is characterized by the net buildup of ECM in the myocardium. Due to the adult mammalian myocardium’s minimal ability for regeneration, diseases that are linked to acute cardiomyocyte mortality are those that produce the most significant fibrotic remodeling of the ventricle. The pathological process of cardiac structural changes and systolic and diastolic dysfunction is brought on by cardiac fibrosis, which is characterized by changes in cardiomyocytes, cardiac fibroblasts, a ratio of collagen (COL) I/III, and an excess production and deposition of ECM, which leads to scar tissue [127].

### 9.2. Kidney Fibrosis

Kidney fibrosis is a defining characteristic of the development and progression of renal lesions in CKD which ultimately leads to ESRD [128]. Glomerulosclerosis and tubulointerstitial fibrosis are two critical stages in the development of CKD. Additionally, in almost all chronic progressive glomerular disorders, the progression of glomerular filtration rate and long-term prognosis is significantly more closely correlated with the degree of tubulointerstitial fibrosis (GFR) than the severity of the glomerular injury [129]. Other significant cellular events in tubule-interstitial fibrosis include the infiltration of inflammatory cells, the activation of fibroblasts and their transformation into myofibroblasts from various sources, the accumulation of a significant amount of ECM components, tubular atrophy, and the rarefaction of peritubular vessels. The majority of monocytes enter the glomerular and interstitial region from the circulation via the peritubular capillary epithelium and infiltrate there when different chemokines and chemoattractants are activated. Damage-causing substances, including ROS and inflammatory and fibrogenic cytokines, are created as a result. Finally, these inflammatory stimuli activate mesangial cells, fibroblasts, and EMT, which results in the production of a substantial portion of ECM components [130]. As demonstrated in Figure 13, TGF-β is crucial in the activation of renal fibrosis. 

### 9.3. Liver Fibrosis

Chronic liver damage is mostly brought on by hepatic fibrosis since it is the main underlying pathology for liver failure and the major characteristic of patients with end-stage liver disease, including hepatocellular carcinoma (HCC) [131]. It can also be caused by alcohol, some medications, autoimmune hepatitis, fatty liver disease, and hepatitis virus infection. It is also a prelude to liver cirrhosis. The primary matrix-producing cells that fuel liver fibrosis are hepatic stellate cells, also known as perisinusoidal cells, which are located in the perisinusoidal space of the liver. Hepatic stellate cell numbers and activation rise along with the production of proinflammatory cytokines and ROS by damaged hepatocytes and stimulated macrophages. HSCs have a significant role in the fibrosis of the liver [132]. As mentioned in the previous section, inflammatory cells such as neutrophils, T lymphocytes, and kupffer cells are drawn to the damaged area and stimulated by DAMPs and ROS to secrete proinflammatory substances such as cytokines (such as interleukin (IL)-1 and IL-6), chemokines (such as C-C chemokine ligand 2 (CC2)), growth factors (such as TGF-β and PDGF), and TNF-α, as discussed in Figure 14 [133]. These profibrotic elements have both direct and indirect effects on quiescent HSCs. Overproducing ECM elements such as collagen, fibronectin, elastin, laminin, and proteoglycan causes activated HSCs to accumulate in the liver [134]. 

### 9.4. Idiopathic Pulmonary Fibrosis (IPF)

Despite the disease’s unclear cause, it is linked to a multitude of illnesses and risk factors, including smoking, farming, and occupational risks, as well as viral and bacterial infections. Excess ECM deposition causes IPF, and it exhibits progressive interstitial fibrosis and interstitial pneumonitis [135]. The architecture of the lung is disrupted by fibroblast and myofibroblast buildup, particularly between the vascular endothelium and the alveolar epithelium, resulting in a “honeycomb” appearance. In IPF patients who have been diagnosed, continuing chemokine and cytokine production suggests that injury and resultant inflammation may still be present [136]. IPF patients have high levels of pro-inflammatory cytokines, such as IL-1, IL-8, IL-18, TNF, MCP-1, and Type-2 cytokines, as well as their receptors, in their BAL fluid or cells taken from a BAL biopsy [137]. All elements of wound healing, including vascular remodeling, myofibroblast differentiation, EMT, TGF-β, and IL-13 production, are significantly impacted by the mixed cytokine profile, which is largely produced by inflammatory cells and leukocytes. Along with their direct fibroblast-activating abilities, TGF-β, and IL-13 have also been found to co-express in IPF, as shown in Figure 15. The diminished apoptotic mechanism expression (Bcl-2 and membrane FAS-L) and fibroblast hyperplasia in IPF can both contribute to the fibrotic response. The culmination of a series of malfunctioning regulatory mechanisms and an excessive cytokine, chemokine, and growth factor release is an unchecked wound-healing response driven by fibroblasts [138].

### 9.5. Muscle Fibrosis

Skeletal muscle fibrosis is regarded as a primary contributing factor to muscular weakness because it negatively impacts muscle regeneration after injury, impairs muscular function, and increases muscle vulnerability to reinjury [139]. Muscular dystrophies, age, and serious muscle injuries are all characterized by skeletal muscle fibrosis. Skeletal muscle fibrosis is a serious clinical issue that might be brought on by many circumstances, including secondary traumatic brain injury or primary skeletal muscle tissue illnesses, such as muscular dystrophy [140]. ECM can accumulate in practically all models of muscle damage or injury, but this is frequently temporary and is assumed to maintain the contractile apparatus while typical regenerative or adaptive mechanisms are active. Contrarily, long-term ECM accumulation impairs function, does not go away under typical physiological circumstances, and is consequently seen as an end-stage process. Skeletal muscle fibrosis could be defined as an abnormal and unresolvable chronic increase of the extracellular connective tissue that interferes with function [141]. Skeletal muscle fibrosis can also be explained in terms of the total amount of collagen in the tissue, which is established by the quantity of hydroxyproline present. This substance is a key component of collagen and is produced when prolyl oxidase hydroxylates the amino acid proline [142]. 

## 10. Novel Antifibrotic Drugs

The understanding that fibrosis is a dynamic and reversible process, improvements in non-invasive fibrosis assessment techniques, and the understanding of the fundamental causes and mediators of fibrosis changes have all contributed to an increased interest in developing effective antifibrotic medications. The therapeutic strategies that are being researched to target various cells and cytokines that encourage fibrosis are covered in the section that follows. There are several points of attack for creating antifibrotic drugs: (1) Eliminate the injury’s underlying causes and any mediators; (2) Lower immunological activity and inflammation; (3) Target-specific signaling includes intracellular signaling and receptor-ligand interactions; (4) Inhibit matrix production and reduce fibrogenesis; (5) Reduce fibrosis by accelerating the breakdown of the scar matrix, inducing stellate cell death, or cell transplantation; (6) Miscellaneous. There are no particular anti-fibrotic medications available today that target fibroblast activation and enhanced ECM production. New antifibrotic methods have recently been researched that utilize specific inhibitors of intracellular tyrosine kinases [139]. New antifibrotic drugs are described in Table 2.

## 11. Conclusions

In summary, fibrosis represents a significant worldwide healthcare burden. Therefore, the identification of critical therapeutic targets highly relevant to human fibrotic disease and the subsequent development of efficient antifibrotic medicines focused on these targets remain research priorities. The complex roles of inflammation and fibrosis in various tissues and disorders have been the topic of a significant number of original and connected contributions to the current research area. In addition, the molecular foundation for fibrosis in organs including the heart, liver, and kidney is being clarified by the ROS/fibrosis paradigm.

## Figures and Tables

**Figure 1 ijms-24-04004-f001:**
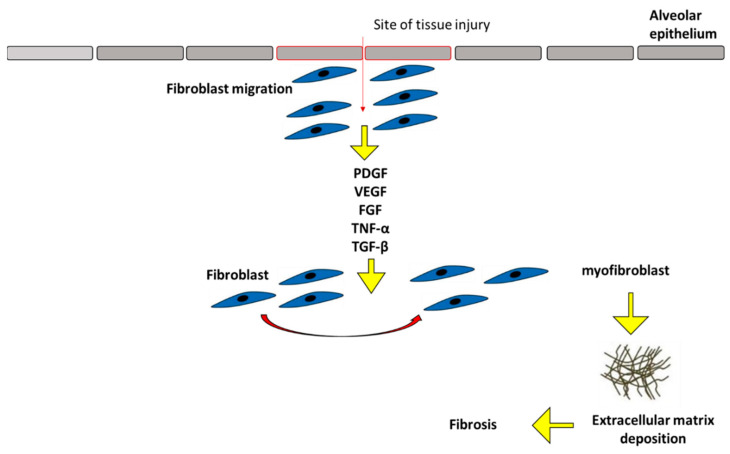
The major mechanisms of fibrosis. PDGF: Platelet-derived growth factor; VEGF: Vascular endothelial growth factor; FGF: Fibroblast growth factor; TNF-α Tumor necrosis factor-alpha; TGF-β: Transforming growth factor beta.

**Figure 2 ijms-24-04004-f002:**
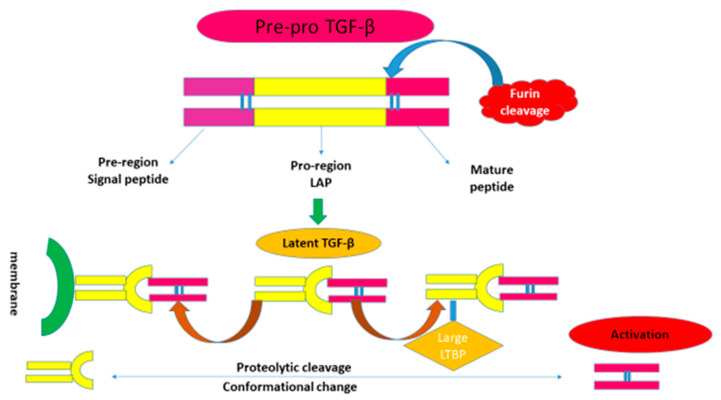
Schematic of the different forms of latent TGF-β. TGF-β: transforming growth factor-beta; LAP: latency-associated peptide. TGF-β is synthesized as an inactive form and cleaved by endopeptidase furin to generate a mature form which is still without biological activity. This is a literature review of the presence of latency-associated peptide (LAP) and latent TGF-β binding protein (LTBP). This large TGF-β associated complex is excreted into the extracellular matrix, cross-linked by tissue transglutaminase, and stored in the tissues in an inactive form. Once TGF-β releases from the latency-maintaining protein complex LAP/LTBP, it will display its powerful biological activity. The active form of TGF-β is a dimer stabilized by hydrophobic interactions and further strengthened by an inter-subunit disulfide bridge in most cases.

**Figure 3 ijms-24-04004-f003:**
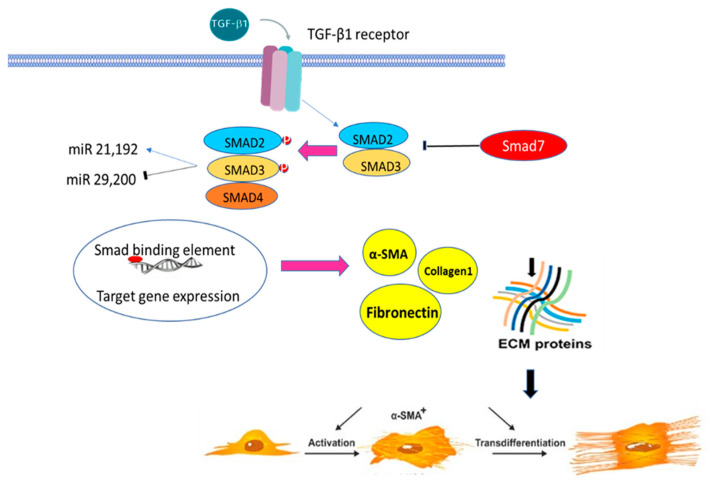
Activation of TGF-β; transforming growth factor-beta, α-SMA; alpha-smooth muscle actin plays a key role in fibrosis through activation of Smad, enhancing ECM deposition and the trans-differentiation of fibroblast to myofibroblast.

**Figure 4 ijms-24-04004-f004:**
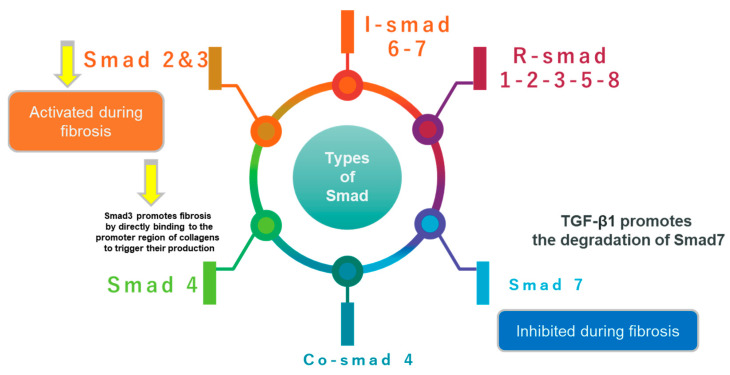
The schematic diagram illustrates the different types of Smad: receptor regulated Smad (R-Smad); common partner-Smad (co-Smad); and Inhibitory Smad (I-Smad).

**Figure 5 ijms-24-04004-f005:**
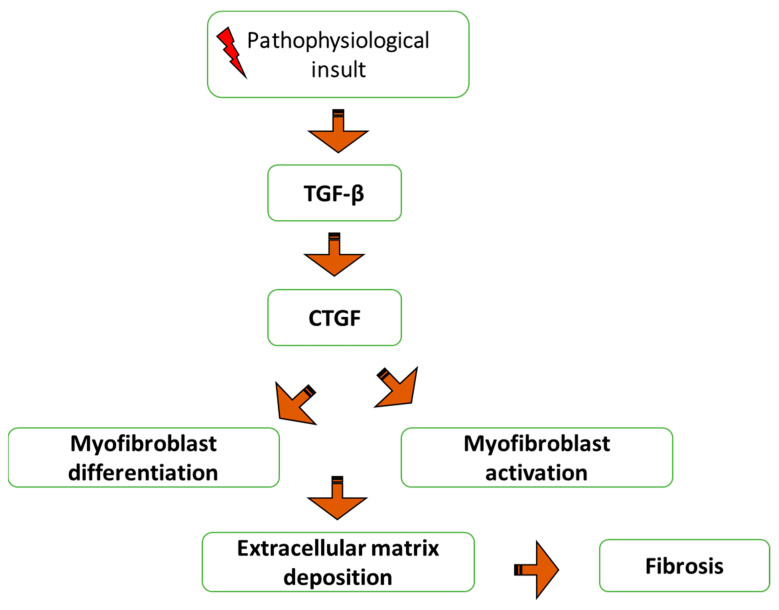
The regulatory scheme for CTGF and its role in fibrosis. TGF-β; transforming growth factor-beta, CTGF; connective tissue growth factor.

**Figure 6 ijms-24-04004-f006:**
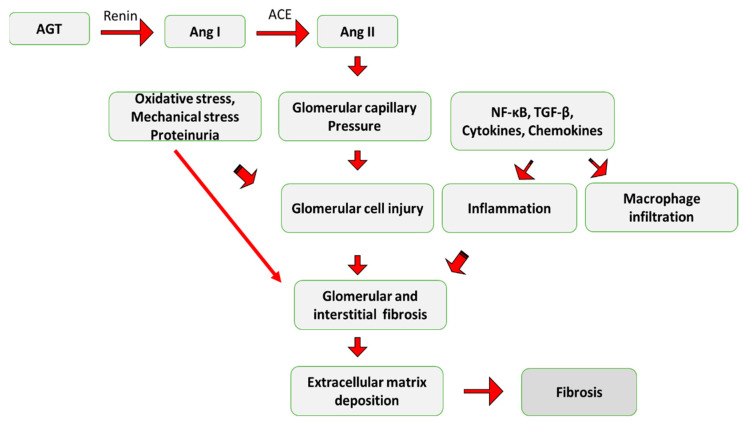
A schematic illustration demonstrates how Angiotensin II (Ang II) contributes to fibrosis. AGT: Angiotensinogen; Ang I: Angiotensin I; ACE: Angiotensin Converting Enzyme.

**Figure 7 ijms-24-04004-f007:**
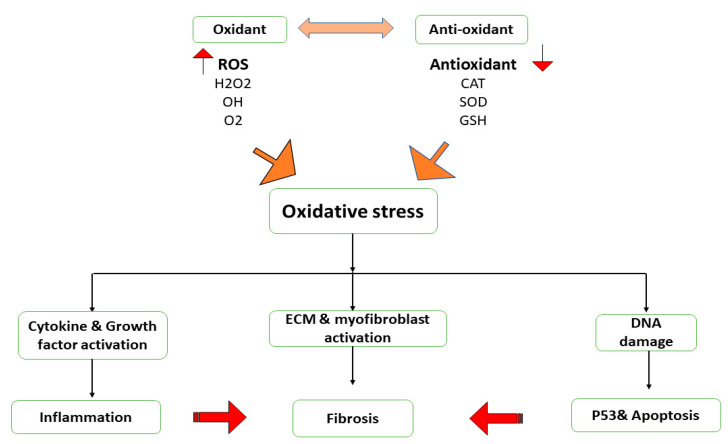
In the pathogenesis of fibrosis, oxidative stress plays a crucial role by promoting inflammation by increasing the production of cytokines and growth factors, increasing myofibroblast differentiation and fibrogenesis, and as a result of DNA damage and p53 activation, ROS promotes apoptosis. These changes aid in the development of fibrosis. CAT; catalase, SOD; superoxide dismutase, GSH; glutathione, H_2_O_2_; Hydrogen peroxide, O_2_: Oxygen.

**Figure 8 ijms-24-04004-f008:**
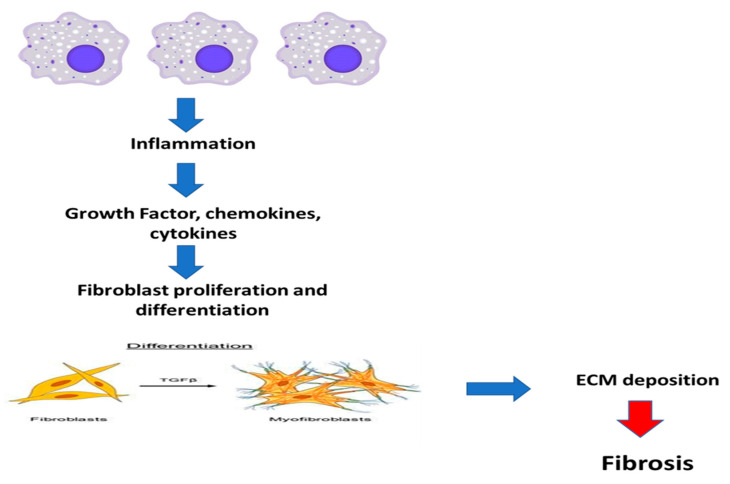
The schematic diagram illustrates how fibrosis is triggered by persistent inflammation. TFG-β; transforming growth factor.

**Figure 9 ijms-24-04004-f009:**
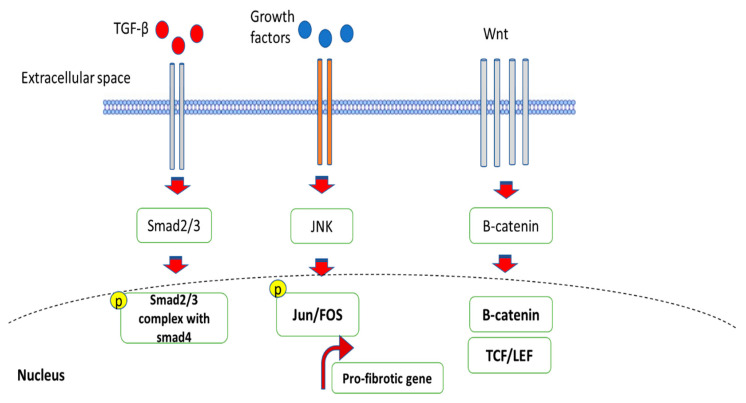
The schematic diagram illustrates the role of Jun N terminal kinase (JNK) in activating the pro-fibrotic genes.

**Figure 10 ijms-24-04004-f010:**
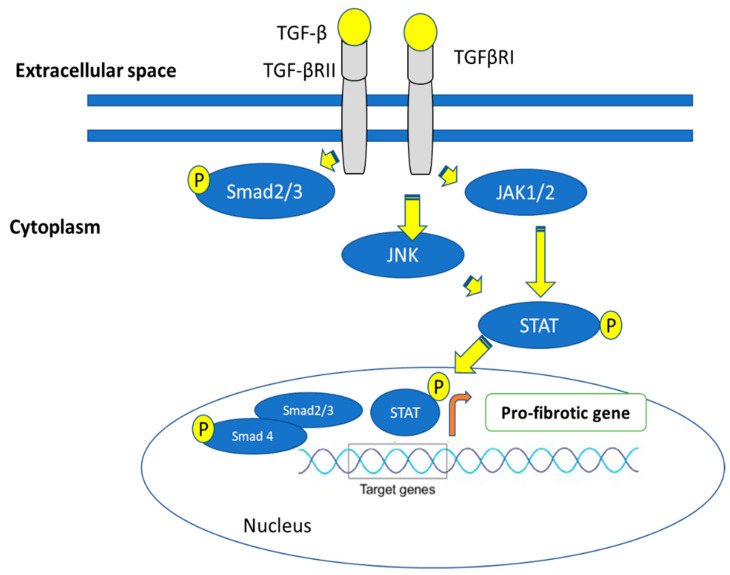
JAK/STAT signaling pathway; When JAK1; Janus kinase connects to TGF-β RI, STAT3; the signal transducer and activator of transcription is activated. Additionally, TGF-β-stimulates STAT3 in a SMAD-dependent way.

**Figure 11 ijms-24-04004-f011:**
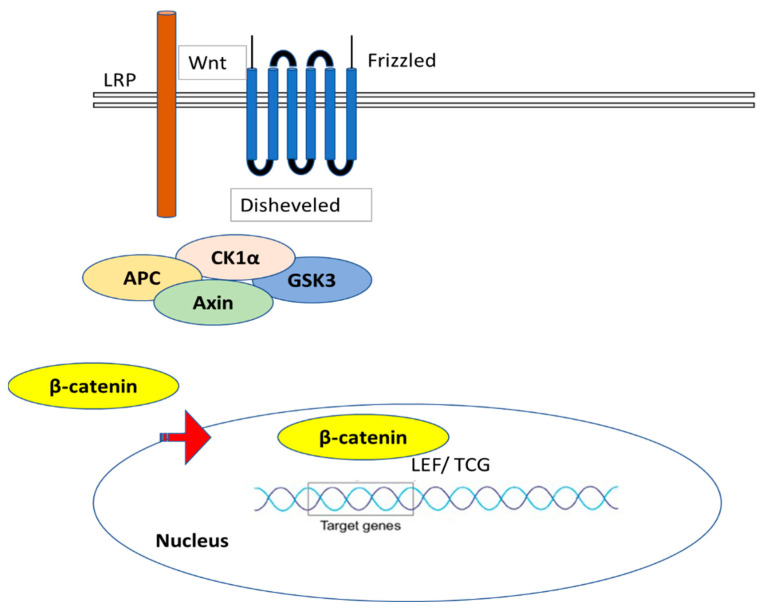
Wnt signaling pathway; Wnt protein interacts with a Frizzled family receptor to initiate Wnt signaling; Axin is taken out of the receptor complex and activates β-catenin to help with receptor activation; the transcription factor on DNA is bound by β-catenin as it travels into the nucleus, activating the transcription of the target genes.

**Figure 12 ijms-24-04004-f012:**
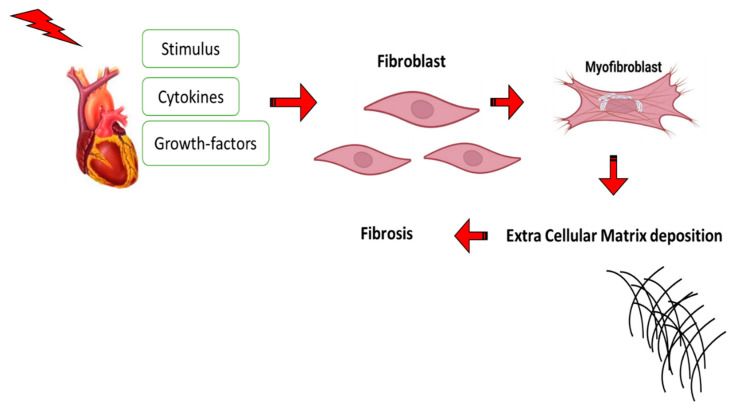
The diagram illustrates cardiac fibrosis; the stimulus encourages the release of cytokines and growth factors, the differentiation of fibroblasts to myofibroblasts, and the deposition of the extracellular matrix.

**Figure 13 ijms-24-04004-f013:**
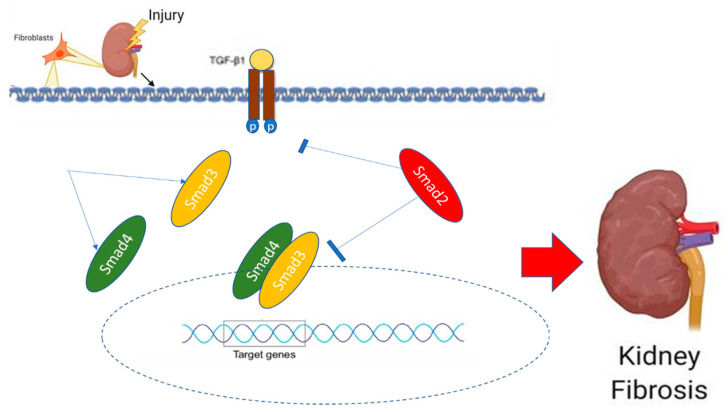
The activation of the Smad2, 3, and 4 complexes by TGF-β cause pro-fibrogenic genes to be induced, which in turn causes the fibrosis process.

**Figure 14 ijms-24-04004-f014:**
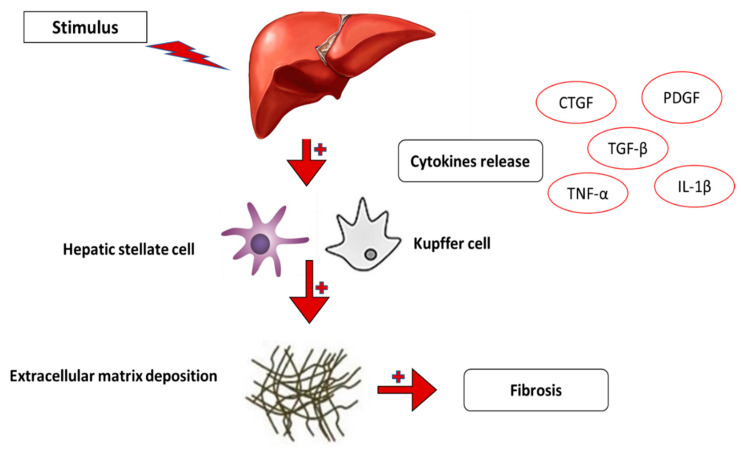
Liver fibrosis results from changes in cytokine release such as connective tissue growth factor (CTGF), transforming growth factor-beta (TGF-β), platelet-derived growth factor (PDGF), tumor necrosis factor-alpha (TNF-α), and the accumulation of extracellular matrix deposition.

**Figure 15 ijms-24-04004-f015:**
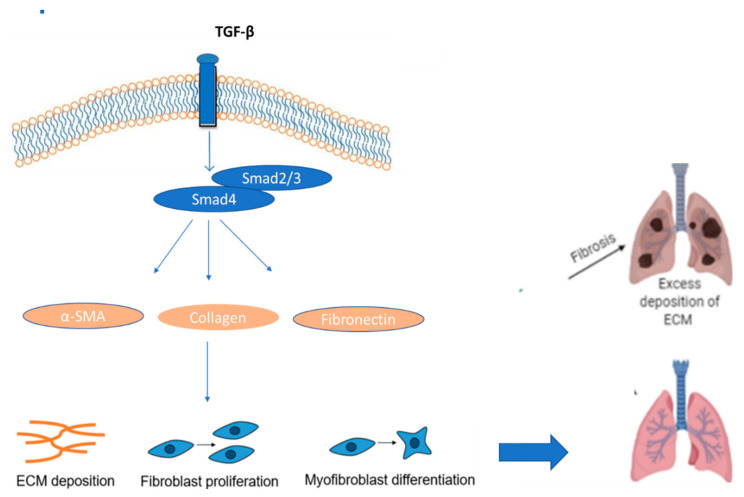
The function of TGF-β in inducing pulmonary fibrosis.

**Table 1 ijms-24-04004-t001:** Antioxidant defense mechanisms.

**Role of Enzymatic Antioxidants**	
Catalase	An H_2_O_2_ scavenger that is expressed in lung AEC and inflammatory cells can stop fibroblasts in IPF lung tissue from becoming activated by H_2_O_2_ [67].
Glutathione (GSH)	One of the best small-molecule antioxidants and one of the most tested indicators is GSH. In mouse models of fibrosis, N-acetyl cysteine (NAC), a precursor to GSH, has been shown to have anti-fibrotic properties. NAC raises lung GSH levels and reduces bleomycin-induced fibrosis [68].
Superoxide dismutase (SOD)	SOD converts superoxide radicals into H_2_O_2_. The distribution and expression of the three mammalian SOD isoforms—intracellular copper-zinc SOD, mitochondrial manganese SOD, and extracellular SOD [EC-SOD]—vary depending on the type of cell [69].
Nuclear factor-erythroid 2-related factor 2 (Nrf2)	The “master regulator” of the antioxidant response is a transcription factor called Nrf2. The antioxidant response element controls hundreds of genes, including NAD(P)H quinone oxidoreductase 1, antioxidant-related genes involved in glutathione biosynthesis, Phase II detoxifying “stress response” genes, and genes regulating the inflammatory and fibrotic responses [70].
**Role of Non-Enzymatic Antioxidants**	
Vitamin-E	As an antioxidant, its primary function is to scavenge loose electrons, or “free radicals”, which can harm cells. Additionally, it strengthens the immune system and prevents heart artery clots from forming [71].
Selenium	This antioxidant assists the body in reducing oxidative stress, which lowers inflammation and improves immunity. Increased blood levels of selenium have been linked to improved immunological response [72].
Vitamin C	It is a hydrophilic free radical scavenger and acts as a reducing and antioxidant agent. Vitamin C and vitamin E interact together synergistically to restore the antioxidant capabilities of oxidized vitamin E, which is necessary for the formation of collagen, carnitine, and neurotransmitters. The antioxidant and prooxidant reserves of ascorbic acid were reported previously [73].
Vitamin A	Because of their ability to scavenge free radicals, carotenoids function as antioxidants. Dietary antioxidants reduce the effectiveness and negative effects of chemotherapy by squelching free radicals and other reactive oxygen species, primarily singlet oxygen species [74].

**Table 2 ijms-24-04004-t002:** Examples of antifibrotic drugs.

Drug	Nature	Mechanism of Action
Nintedanib	Tyrosine kinase inhibitor	It is an efficient and well-tolerated tyrosine kinase inhibitor (TKI) that showed an important anti-fibrotic effect in patients with chronic graft-versus-host disease (GVHD). It is also used to treat idiopathic pulmonary fibrosis [143]. Focusing on upstream receptors necessary for the growth of fibrosis inhibits the proliferation, migration, and transformation of fibroblasts. It blocks the fibroblast growth factor receptor, vascular endothelial growth factor receptor, and platelet-derived growth factor receptor binding sites [144].
Pirfenidone	Orally active (modified phenyl pyridine)	It is used to treat idiopathic pulmonary fibrosis and inhibits the production and activity of TGF-β. It can diminish fibroblast proliferation and inhibit collagen formation as well as the transcription of the TGF-1 gene and the expression of collagen type 1 mRNA [16].
Imatinib mesylate	Tyrosine kinase inhibitor	It prevents the progression of fibrosis in systemic sclerosis patients and is used in the treatment of established fibrosis. It binds to the Abelson kinase (c-AblATP-binding) site and effectively inhibits its tyrosine kinase activity, which necessitates the conversion of ATP into ADP and the phosphorylation of target proteins. An essential TGF- and PDGF downstream signaling molecule is c-Abl [145].
Halofuginone	Plant alkaloid (from *Dichroa febrifuga*)	It is used on patients with cutaneous cGvHD, a condition marked by significant skin fibrosis and contractures. It leads to skin integrity decline and a dose-dependent reduction in the skin’s collagen content [146].
Relaxin	A polypeptide of the insulin/relaxin superfamily	It has antifibrotic effects in experimental models of renal fibrosis. It reduces collagen synthesis and encourages collagen breakdown by raising MMP levels and activity. A lot of relaxin’s effects are antagonistic to TGF’s effects. Relaxin’s receptor (RXFP1) was only recently discovered, which accelerated the development of novel antifibrotic relaxin discoveries. The effects of relaxin as an antifibrotic drug in cardiac, liver, kidney, lung and even cutaneous fibrosis are demonstrated by clinical investigations [147].
Oltipraz	Cancer chemo-preventive agent	It is used for treatment in patients with liver fibrosis and cirrhosis. The inhibition of matrix synthesis and the number of target cytokines engaged in this process is increasing. Since TGF-β1 is the most powerful inducer of collagen I and other matrix components, blocking its effects continues to be a primary goal of liver antifibrotic initiatives [148].

## Data Availability

The data presented in this study are available inside the manuscript.

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
