# Peer review of "Fibrosis: Types, Effects, Markers, Mechanisms for Disease Progression, and Its Relation with Oxidative Stress, Immunity, and Inflammation"

_ijms, 2023, doi:10.3390/ijms24044004_

Round 1

Reviewer 1 Report

The manuscript is a well written review. It summarize the mechanism of tissue fibrosis in different aspects, including immunity, inflammation and oxidative stress. However, some details should be improved.

1) Introducing the function of TGF-β and smad in fibrosis, TGF-β1, TGF-β2 or TGF-β3 should be indicated clearly. Such as line 110, we do not know which form of TGF-β function. In addition, the tissue should also be indicated clearly, but not just fibrosis.

2) In paragraph-Antioxidant, the function of antioxidant on fibrosis and mechanism should be introduced in detail. Or, there is no enough evidences to support it.

3) I suggest that immunity can be added into the title.

4)Wnt pathway and JAK-STAT signal should not be coordinated with immunity ,oxidative stress and inflammation

Line 68   line 157  β is missing

Line 234  what is IPF

Line 278 α is missing

Lin3 36 TGF?

Author Response

                                                       Reviewer #1

Comments and Suggestions for Authors

The manuscript is a well written review. It summarize the mechanism of tissue fibrosis in different aspects, including immunity, inflammation and oxidative stress. However, some details should be improved.

Answer: The authors thank the reviewer for his effort to improve the quality of our manuscript. All the reviewer’s comments were amended accordingly.

1) Introducing the function of TGF-β and smad in fibrosis, TGF-β1, TGF-β2 or TGF-β3 should be indicated clearly. Such as line 110, we do not know which form of TGF-β function. In addition, the tissue should also be indicated clearly, but not just fibrosis.

Answer: Ok, the function of TGF-β and smad in fibrosis, TGF-β1, TGF-β2 or TGF-β3 were discussed according to the reviewer’s suggestions.  

2) In paragraph-Antioxidant, the function of antioxidant on fibrosis and mechanism should be introduced in detail. Or, there is no enough evidences to support it.

Answer: Ok, the functions of antioxidants on fibrosis and their mechanisms were introduced in detail as requested.  

3) I suggest that immunity can be added into the title.

Answer: Ok, added as requested.

4) Wnt pathway and JAK-STAT signal should not be coordinated with immunity ,oxidative stress and inflammation.

Answer: Thanks for your comment. However, the authors disagree with the respected reviewer for his opinion for the following reasons:

1. Relation between WNT Pathway and oxidative stress:

The metabolically generated H2O2 has emerged as a central hub in redox signaling and oxidative stress. Malignant cancer cells are known to contain and tolerate higher ROS levels than normal cells, which is due to distorted cellular metabolism. A recent study indicated sthat CRC cells, even when deprived of growth factors under acute oxidative distress conditions by H2O2, promote β-Catenin expression and modulate APC protein. In addition, H2O2 induces a differential gene expression in the analyzed models, depending on the tumor phenotype and leading to favoring both WNT/β–Catenin-dependent and -independent signaling [1]. Also, oxidative stress upregulates Wnt signaling in human retinal microvascular endothelial cells through activation of disheveled [2].

2. Activation of canonical Wnt signalling is required for TGF-β-mediated fibrosis:

The activation of the canonical Wnt pathway has a key role for fibroblast activation and collagen release in fibrosis. Wnt signalling stimulated the differentiation of resting fibroblasts into myofibroblasts, increased the release of extracellular matrix components and induced fibrosis [3].

3. JAK inhibitors and oxidative stress control:

The STAT family of transcription factors is known to be activated by most cytokines and growth factors, we have shown that, in addition to these agonists, STAT family members can also be activated by oxidative stress.  Primary Sjögren's syndrome (SjS) is a complex autoimmune epithelitis, with few treatment options, but the use of Janus kinase (JAK) inhibitors is promising because suppression of the JAK/signal transducer and activator of transcription (STAT) pathway improves sicca manifestations. Playing a primary and pathogenic role in disease development, the oxidative stress response is upregulated in activated salivary gland epithelial cells (SGECs) from patients with SjS [4]. 

4. JAK-STAT signalling and the atrial fibrillation promoting fibrotic substrate:

JAK-STAT inhibition reduces the profibrotic effects of PDGF stimulation on canine fibroblasts in vitro while attenuating in vivo LA-fibrosis and remodelling in post-MI mice, suggesting that the JAK/STAT pathway contributes to LA-fibrogenesis and might be a potential target for LA-fibrosis prevention [5].

Line 68   line 157  β is missing

Answer: Ok, corrected.

Line 234  what is IPF

Answer: IPF is Idiopathic Pulmonary Fibrosis, added as requested.

Line 278 α is missing

Answer: Ok, corrected.

Lin3 36 TGF?

Answer: Ok, done.

References

  1. Catalano, T.; D’Amico, E.; Moscatello, C.; Di Marcantonio, M.C.; Ferrone, A.; Bologna, G.; Selvaggi, F.; Lanuti, P.; Cotellese, R.; Curia, M.C. Oxidative Distress Induces Wnt/β-Catenin Pathway Modulation in Colorectal Cancer Cells: Perspectives on APC Retained Functions. Cancers 2021, 13, 6045.
  2. Zhang, C.; Tannous, E.; Zheng, J.J. Oxidative stress upregulates Wnt signaling in human retinal microvascular endothelial cells through activation of disheveled. Journal of cellular biochemistry 2019, 120, 14044-14054.
  3. Akhmetshina, A.; Palumbo, K.; Dees, C.; Bergmann, C.; Venalis, P.; Zerr, P.; Horn, A.; Kireva, T.; Beyer, C.; Zwerina, J. Activation of canonical Wnt signalling is required for TGF-β-mediated fibrosis. Nature communications 2012, 3, 1-12.
  4. Charras, A.; Arvaniti, P.; Le Dantec, C.; Dalekos, G.N.; Zachou, K.; Bordron, A.; Renaudineau, Y. JAK inhibitors and oxidative stress control. Frontiers in Immunology 2019, 10, 2814.
  5. Chen, Y.; Surinkaew, S.; Naud, P.; Qi, X.-Y.; Gillis, M.-A.; Shi, Y.-F.; Tardif, J.-C.; Dobrev, D.; Nattel, S. JAK-STAT signalling and the atrial fibrillation promoting fibrotic substrate. Cardiovascular research 2017, 113, 310-320.

Reviewer 2 Report

This is a nice review of the various pathways that contribute to fibrosis. It is very broad ranging, and a great overview.  I have only minor comments

Line

48 please explain ‘tubular cells’

48-49 sounds like all mesenchymal cells are monocytes, please recheck and rephrase

Throughout – search on “It has been shown that’ and similar phrases and delete these to reduce the wordiness: for example change things like

new research has shown that Smad3 causes fibrosis

to

Smad3 causes fibrosis

144 “CTGF is secreted, and its biology is an important expression.” Doesn’t make sense – please rephrase

203 ‘circular profibrogenic circle’ change to ‘positive feedback loop’

225 and on – please double check this – I was under the impression that antioxidants have not worked well in IPF trials

301 ‘TIMP1 is a pro-fibrotic factor’ seems to be in the wrong sentence

332 beta may be missing

435 please explain / clarify ‘a ratio of collagen (COL) I/III,’

Please recheck the abbreviations list, for instance I could not find ‘ESRD’

Author Response

                                                   Reviewer #2

Comments and Suggestions for Authors

This is a nice review of the various pathways that contribute to fibrosis. It is very broad ranging, and a great overview. I have only minor comments.

Answer: The authors thank the reviewer for his great effort to improve the quality of our manuscript.

All the reviewer’s comments were amended accordingly.

Line

48 please explain ‘tubular cells’

Answer: Thanks for your observation. It was a mistake and was corrected.

48-49 sounds like all mesenchymal cells are monocytes, please recheck and rephrase

Answer: Ok, rechecked as requested.

Throughout – search on “It has been shown that’ and similar phrases and delete these to reduce the wordiness: for example change things like new research has shown that Smad3 causes fibrosis to Smad3 causes fibrosis.

Answer: Thanks for your suggestion. These sentences were modified accordingly.

 144 “CTGF is secreted, and its biology is an important expression.” Doesn’t make sense – please rephrase.

Answer: Ok, rephrased as requested.

203 ‘circular profibrogenic circle’ change to ‘positive feedback loop’.

Answer: Ok, changed as requested.

 225 and on – please double check this – I was under the impression that antioxidants have not worked well in IPF trials.

Answer: Ok, done.

301 ‘TIMP1 is a pro-fibrotic factor’ seems to be in the wrong sentence.

Answer: Thanks, it was corrected.

332 beta may be missing.

Answer: Ok, added.

435 please explain/ clarify ‘a ratio of collagen (COL) I/III,’

Answer: There are at least 14 different types of collagen, type I and III are the two predominant types implicated in patients with connective tissue disorders, like diverticulosis. Type I collagen is composed of rigid fibrils and is the predominant type found in the body. Type III collagen is thinner and generally regarded as immature and weak. The typical ratio of type I to III collagen is four to one [1]. For example, in hernia formation and aortic dissection collagen defects with a higher rate of elastic type III collagen, have been found.

Please recheck the abbreviations list, for instance I could not find ‘ESRD’

Answer: Ok, the abbreviations list was rechecked and modified as suggested.

Reference

1. Stumpf, M.; Cao, W.; Klinge, U.; Klosterhalfen, B.; Kasperk, R.; Schumpelick, V.J.I.j.o.c.d. Increased distribution of collagen type III and reduced expression of matrix metalloproteinase 1 in patients with diverticular disease. 2001, 16, 271-275.

Reviewer 3 Report

Fibrosis is a huge topic, the authors tried to highlight some basic mechanisms of the disease. They tried to include figures in many sections to help to guide the reading, which effort is appreciated.

Due to the complicated mechanisms and causes of fibrosis, it is understandable for the authors to simplify the content. However, sometimes it feels it is over-simplified, and make this an introductory guide for fibrotic disease.  It did not give many new and updated information. Also, many places are mixing the terms of fibroblast and myofibroblast. These are very important concepts in fibrosis. Normally, fibroblast is activated to become myofibroblast, fibroblast differentiate into myofibroblast. While myofibroblast should not be differentiate further. There are several places have mixed up these two terms. One place is on top of p.8.

Many of the figures only have brief description, which make it hard to understand what are trying to explain. For example, Figure 2 is very confusing. Should have explained what are those half circles of green and yellow, what is the process are described.

Most of the figures are significantly simplified, which makes many of these resemble others.   Some of the figures simplified too much, like Figure 9.

Even many fibrosis diseases have similar process, but this review did not try to present each organ's unique characteristics. In the mechanistic part, some mechanisms only fit some fibrosis, not all. Thus, when describe certain mechanism, should mention which fibrosis that mechanism applies to.  Similarly, in Table 2 for antifirbotic drugs, should list the drug is used for which fibrosis.

There are some minor mistakes, such as p.13, the subtitle is 4.1, while p.12 numbered 5, and p. 15 numbered 5 again.

Additionally, cystic fibrosis is not a fibrotic disease, even it has fibrosis in the disease name.

Some of the wording should be changed, and use more widely used terms. For example, p.2, "pro-fibrotic chemicals" people usually use as "pro-fibrotic factors and cytokines";  p.4 "It is widely know Smad3 is harmful" is not accurate; and p.14 "The kidney fibroblasts are encouraged to develop a myofibroblast phenotype by enhancing the production of a-SMA", which is confusing. 

Author Response

                                                     Reviewer #3

Comments and Suggestions for Authors

Fibrosis is a huge topic, the authors tried to highlight some basic mechanisms of the disease. They tried to include figures in many sections to help to guide the reading, which effort is appreciated.

Due to the complicated mechanisms and causes of fibrosis, it is understandable for the authors to simplify the content. However, sometimes it feels it is over-simplified, and make this an introductory guide for fibrotic disease.  It did not give many new and updated information. Also, many places are mixing the terms of fibroblast and myofibroblast. These are very important concepts in fibrosis. Normally, fibroblast is activated to become myofibroblast, fibroblast differentiate into myofibroblast. While myofibroblast should not be differentiate further. There are several places have mixed up these two terms. One place is on top of p.8.

Answer: The authors thank the reviewer for his great effort to improve the quality of our manuscript.

As the respected reviewer said that the mechanisms of fibrosis are complicated to be understood, so the authors tried to simplify these mechanisms as possible in a scientific manner. This was done to help other scientists all over the world working on fibrosis or even are interested in fibrosis to understand these mechanisms as possible. Also, the authors intended to cover the main aspects and sections of fibrosis as possible. However, the reviewer’s recommendations were amended with great interest. Regards.

Many of the figures only have brief description, which make it hard to understand what are trying to explain. For example, Figure 2 is very confusing. Should have explained what are those half circles of green and yellow, what is the process are described.

Most of the figures are significantly simplified, which makes many of these resemble others.   Some of the figures simplified too much, like Figure 9.

Answer: Ok, Figure 2 was modified according to the reviewer’s comment. Regarding Figure 9, it was aforementioned that the authors intended to simplify the fibrosis mechanisms as possible in a scientific manner which is a strong point in our review. Thanks in advance.

Even many fibrosis diseases have similar process, but this review did not try to present each organ's unique characteristics. In the mechanistic part, some mechanisms only fit some fibrosis, not all. Thus, when describe certain mechanism, should mention which fibrosis that mechanism applies to.  Similarly, in Table 2 for antifirbotic drugs, should list the drug is used for which fibrosis.

Answer: The authors thank the reviewer for his recommendations. All the requested modifications were done accordingly.

There are some minor mistakes, such as p.13, the subtitle is 4.1, while p.12 numbered 5, and p. 15 numbered 5 again.

Answer: Ok, corrected.

Additionally, cystic fibrosis is not a fibrotic disease, even it has fibrosis in the disease name.

Answer: Ok, modified as requested.

Some of the wording should be changed, and use more widely used terms. For example, p.2, "pro-fibrotic chemicals" people usually use as "pro-fibrotic factors and cytokines";  p.4 "It is widely know Smad3 is harmful" is not accurate; and p.14 "The kidney fibroblasts are encouraged to develop a myofibroblast phenotype by enhancing the production of a-SMA", which is confusing.

Answer: Ok, all the requested modifications were done. Moreover, the manuscript was revised thoroughly again for any grammatical and/or typos errors. Thanks in advance.

Reviewer 4 Report

the manuscript would be very interesting because the topic treated by the authors is involved in many patological diseases. However, many points are treated only superficially and deserve to be improved. In particular:

Line 64-65: TGFB has also a key role in inducing epithelial-mesenchymal transition (EMT). In fact, it can downregulate cell junctions  favoring cell proliferation and survival  in inflammatory deseases (PMID: 26739007,26708185)

2.5. Role of nuclear erythroid 2-related factor 2 (Nrf2) in fibrosis: The role of NRF2/KEAP1 signalling deserves to be better introduced. In particular, It deserves to be pointed out that this signalling is involved also in cancer progression and chemoresistance. Moreover, modulatingthe expression of important antioxidant enzymes plays a key role in cancer prevention (as also recently reviewed PMID: 36335520, 35901941)  

Line 148: define EMT

9. Novel antifibrotic drugs: In this paragraph deserves to be mentioned Nilotinib, an efficient and well tollerated Tyrosine kinase inhibitors (TKIs) which showed an important anti-fibrotic effect in patients with chronic graft-versus-host disease (GVHD) (see PMID: 32006713)

An accurate revision of typing errors is recommended

The list of abbreviations in the table must be written in alphabetical order

the presence of paragraph composed by only few sentences must be avoid (see for example lines 330-333)

Author Response

                                               Reviewer #4

Comments and Suggestions for Authors

the manuscript would be very interesting because the topic treated by the authors is involved in many patological diseases. However, many points are treated only superficially and deserve to be improved. In particular:

Answer: The authors thank the reviewer for his great effort to improve the quality of our manuscript.

All the recommended modifications were done as requested.

Line 64-65: TGFB has also a key role in inducing epithelial-mesenchymal transition (EMT). In fact, it can downregulate cell junctions  favoring cell proliferation and survival  in inflammatory deseases (PMID: 26739007,26708185).

Answer: Ok, the suggested topic was discussed using the recommended references.

2.5. Role of nuclear erythroid 2-related factor 2 (Nrf2) in fibrosis: The role of NRF2/KEAP1 signalling deserves to be better introduced. In particular, It deserves to be pointed out that this signalling is involved also in cancer progression and chemoresistance. Moreover, modulatingthe expression of important antioxidant enzymes plays a key role in cancer prevention (as also recently reviewed PMID: 36335520, 35901941).

  Answer: Ok, the suggested topic was discussed using the recommended references.

Line 148: define EMT.

Answer: Ok, defined as requested.

  1. Novel antifibrotic drugs: In this paragraph deserves to be mentioned Nilotinib, an efficient and well tollerated Tyrosine kinase inhibitors (TKIs) which showed an important anti-fibrotic effect in patients with chronic graft-versus-host disease (GVHD) (see PMID: 32006713).

Answer: Ok, the suggested drug was mentioned and discussed using the recommended references.

An accurate revision of typing errors is recommended.

Answer: Ok, the manuscript was revised thoroughly again for any grammatical and/or typos errors.

The list of abbreviations in the table must be written in alphabetical order.

Answer: Ok, the list of abbreviations in the table was written in alphabetical order as requested.

The presence of paragraph composed by only few sentences must be avoid (see for example lines 330-333)

Answer: Ok, done as requested.

Round 2

Reviewer 1 Report

The questions were well addressed.

Reviewer 2 Report

Nice job, nice review

Reviewer 4 Report

The manuscript has been significantly improved and can be accepted in the present form. However, the reference style should be reviewed